# Photonic time-crystalline behaviour mediated by phonon squeezing in *Ta₂NiSe₅*

**Marios H. Michael** [1,2] ✉, **Sheikh Rubaiat Ul Haque** [3,4] ✉, **Lukas Windgaetter²**, **Simone Latini²**, **Yuan Zhang³**, **Angel Rubio** [2,5], **Richard D. Averitt** [3] & **Eugene Demler¹,⁶**

Photonic time crystals refer to materials whose dielectric properties are periodic in time, analogous to a photonic crystal whose dielectric properties is periodic in space. Here, we theoretically investigate photonic time-crystalline behaviour initiated by optical excitation above the electronic gap of the excitonic insulator candidate Ta₂NiSe₅. We show that after electron photo-excitation, electron-phonon coupling leads to an unconventional squeezed phonon state, characterised by periodic oscillations of phonon fluctuations. Squeezing oscillations lead to photonic time crystalline behaviour. The key signature of the photonic time crystalline behaviour is terahertz (THz) amplification of reflectivity in a narrow frequency band. The theory is supported by experimental results on Ta₂NiSe₅ where photoexcitation with short pulses leads to enhanced THz reflectivity with the predicted features. We explain the key mechanism leading to THz amplification in terms of a simplified electron-phonon Hamiltonian motivated by ab-initio DFT calculations. Our theory suggests that the pumped Ta₂NiSe₅ is a gain medium, demonstrating that squeezed phonon noise may be used to create THz amplifiers in THz communication applications.

Manipulating materials in time and space through optical pumping paves the way for new phenomena and functionalities. A notable example in this field is *photonic time crystals*, defined as materials whose effective dielectric properties oscillate periodically in time, in analogy to photonic crystals whose effective dielectric properties oscillate periodically in space (Photonic time crystals should not be confused with *discrete time crystals* that refer to spontaneous symmetry-breaking in time[1].). Photonic time-crystals have emerged as a promising platform for terahertz (THz) amplification, lasing[2,3], or diffraction scattering in time[4], with potential applications in creating new THz lasers and enabling 6G wireless technology[5].

An effective way of creating such photonic time crystals is by exciting the collective dynamics of strongly correlated quantum materials in pump and probe experiments. This assertion is supported by several experiments that have demonstrated dramatic changes in the THz and mid-IR reflectivity following photoexcitation. In experiments on YBa₂Cu₃O₆.₅ cuprate superconductors, exciting Josephson plasma oscillations induce extra "edge" features in the optical reflectivity[6–9]. On the other hand, the phonon-polariton system S*iC*[10] and the bulk superconductor K₃C₆₀[11,12] provide examples where oscillations inside these materials result in significant reflectivity enhancement, possibly exceeding unity. A unified interpretation of these seemingly disparate pump-induced features can be formulated in terms of Floquet theory under the assumption that collective excitations effectively create a photonic time crystal for the duration of their lifetime.

¹Department of Physics, Harvard University, Cambridge, MA 02138, USA. ²Max Planck Institute for the Structure and Dynamics of Matter, Luruper Chausse 149, 22761 Hamburg, Germany. ³Department of Physics, University of California San Diego, La Jolla, CA 92093, USA. ⁴Stanford Institute for Materials and Energy Sciences, SLAC National Accelerator Laboratory, Menlo Park, CA 94025, USA. ⁵Center for Computational Quantum Physics, The Flatiron Institute, 162 Fifth Avenue, New York 10010 NY, USA. ⁶Institute for Theoretical Physics, ETH Zürich, 8093 Zürich, Switzerland. ✉e-mail: marios.michael@mpsd.mpg.de; rubaiath@stanford.edu

Previous analysis has demonstrated the existence of four types of drive-induced features in reflectivity, depending on the relative strength of parametric driving and dissipation[2]. Edge-like features occur from interference between different Floquet components of the transmission channels when dissipation is strong compared to the oscillation amplitude. In the opposite regime, strong amplification of reflectivity occurs when the parametric drive is not compensated by dissipation and the material exhibits a lasing instability. In fact, such reflectivity features can serve as reporters of a lasing instability, indicating that the effective Floquet medium can be used as a gain medium in a laser.

In this article, we provide a theory that exemplifies that THz noise of phonons can be used to create a THz photonic time crystal in the correlated material $Ta_2NiSe_5$. In particular, we are able to explain pump-induced THz amplification of reflectivity in $Ta_2NiSe_5$ that is observed experimentally.

$Ta_2NiSe_5$ attracted considerable attention in the literature due to its potential as an excitonic insulator candidate[13–22]. This material has an insulating gap below the critical temperature which grows to be 160 meV at low temperatures. The pump-probe experiment demonstrates that an optical excitation at frequencies above the electronic gap induces reflectivity amplification which is maximal at 4.7 THz; a frequency that corresponds to an IR-active $B_{3u}$ phonon mode seen previously in infrared measurements[19,20]. This turns out to be a surprising result given that the high-frequency pump cannot directly excite the collective THz modes. Instead, we show phenomenologically and using ab initio calculations that this dramatic down conversion is the consequence of creating a photonic time crystal through phonon squeezing.

The multi-step process described in this article, converting high-frequency pumping to THz amplification of reflectivity, is schematically illustrated in Fig. 1a and is outlined as follows: (a) The pump excites electrons from the valence bands to the conduction bands through direct dipole transitions. (b) The photoexcited electrons exhibit an unusual quadratic coupling to the 4.7 THz phonons (see Results):

$$H_{el-ph} = \sum_k g_k n_{el,k} Q^2,\tag{1}$$

where $Q$ is the IR-active phonon displacement, $n_{el,k}$ is the photoexcited electron occupation, and $g_k$ is the effective electron-phonon coupling. The pump is non-resonant with the IR-active THz phonons and thus does not directly initiate the enhanced reflectivity dynamics. Instead, IR-active phonon pair generation occurs via a Raman process caused by pump-induced changes in the electronic occupation. The result is that even though the expectation value of the phonon displacement is zero, $\langle Q \rangle = 0$, the fluctuations are squeezed and coherently oscillate at twice the phonon frequency, $\langle Q^2(t) \rangle = \langle Q^2 \rangle_0 + A\cos(2\omega_{ph}t)$. (c) The squeezed phonon oscillations and phonon nonlinearities create a Floquet material/photonic time crystal oscillating at $2\omega_{ph} = 9.4$ THz. d) Parametric resonances due to the oscillating field occur primarily around $\omega_{ph}$, parametrically amplifying the reflectivity as depicted in Fig. 1b.

The parametric amplification of reflectivity in the photonic time crystal created by the squeezing phonon oscillations can be understood as follows. The oscillations of phonon fluctuations, parametrically drive the rest of the material through nonlinear interactions, which can be schematically represented as:

$$H_{ampl.} = X(t)a^\dagger_{1,k}a^\dagger_{2,-k} + h.c.,\tag{2}$$

**Fig. 1 | Schematic mechanism of time crystalline behaviour that leads to amplification of THz optical reflectivity following high-frequency pumping.**
**a** An ultrafast laser pulse (0.5eV, 45 fs) photoexcites electrons through direct dipole transitions between the valence and conduction bands. The photoexcited electrons create a squeezed state of phonons through a strong electron-phonon interaction (see equation (1)). **b** Reflectivity amplification of pumped $Ta_2NiSe_5$. Fluctuations of phonons in the squeezed state oscillate at twice the phonon frequency, $\omega_{ph}$, creating an effective THz Floquet medium/ Photonic time crystal. Parametric driving from the phonon oscillations can create pairs of photons at the signal and idler frequency once stimulated by the probe pulse. This enhances the reflectivity and also scatters counter-propagating light oscillating at the idler frequency.
**c** Relative change in the reflectivity as a function of frequency subsequent to photoexcitation. The experimental results from[23] are shown at two different temperatures together with the theoretical fit which considers parametric amplification by a 9.4 THz oscillating field.

where $X(t)$ is the squeezing-induced drive while $\{a_{1,k}^\dagger, a_{2,-k}^\dagger\}$ are the creation operators of photons with opposite momentum which may be associated with different phonon-polariton bands {1, 2} at frequencies parametrically resonant with the drive, $\omega_1(k) + \omega_2(-k) = \omega_d$. As shown in Fig. 1b, the Floquet drive can generate pairs of photons at the signal frequency of the incoming probe, $\omega_s$, and a photon oscillating at the idler frequency, $\omega_{id} = \omega_d - \omega_s$. Reflectivity amplification is then caused by stimulated emission of photon pairs when probing the driven material. To compute the reflection coefficient, we use degenerate Floquet perturbation theory to construct the eigenmodes and solve the Fresnel equations at the boundary.

Our analysis shows that the parametrically amplified reflectivity at 4.7 THz is consistent with the creation of a photonic time crystal whose parameters oscillate at 9.4 THz. The orange and green bold lines in Fig. 1c depict experimental results of photoinduced reflectivity enhancement of T$a_2$NiSe$_5$ single crystals obtained by time-resolved broadband THz spectroscopy as a function of frequency at temperatures of 90 K and 120 K, respectively (for details on the experimental setup, see Methods and the accompanying paper in ref. 23). Comparing the experimental results with the theoretical fit as plotted in Fig. 1c, it can be seen that the theoretical fit captures the experimental amplification profile with reasonable accuracy.

We then proceed to investigate the origin of the 9.4 THz oscillation. We show that electron-phonon coupling between electronic bands involved in the photoexcitation and the 4.7 THz IR-phonon naturally leads to phonon squeezing. Phonon squeezing oscillations have twice the frequency of the phonon and through nonlinear interactions lead to a photonic time crystal at twice the phonon frequency. We find that this mechanism is generic and should apply to all IR-phonons present in the system. However, for phonon squeezing oscillations to create an effective Floquet material that can significantly enhance the reflectivity, the phonon needs to be both strongly coupled to the photoexcited electronic bands and also have a strong coupling to the electric field.

We hence perform ab initio calculations to determine both the magnitude of the electron-phonon coupling and the IR activity of the different phonon modes. We find that, indeed, the 4.7 THz IR-phonon is special, exhibiting strong electron-phonon coupling and appreciable IR activity as compared to other IR-phonons in the same frequency range. The theory shows that the predicted reflectivity amplification is a reporter of a lasing instability, implying that pumped T$a_2$NiSe$_5$ could, in principle, serve as a gain medium for THz lasing.

As a final intriguing aspect, we find that the 4.7 THz mode is strongly coupled to the elusive and hotly debated order parameter of T$a_2$NiSe$_5$[14,16,17,24–27], which is thought to have some excitonic insulating character. A number of state-of-the-art experiments have been performed to shed light on the origin of the phase transition at 326 K as well as what role the excitonic and lattice instabilities play in this phase transition process. However, the results show contradictory results[15,18–22,24,28–43] and the question of whether T$a_2$NiSe$_5$ hosts an excitonic insulator phase remains an open question. Using DFT frozen phonon calculations, we find that the electron-phonon coupling for the 4.7 THz phonon effectively vanishes in the high-temperature orthorhombic phase. This indicates that the THz parametric amplification of reflectivity is mediated by the low-temperature phase and is expected to be sensitive to the phase transition. The interplay between the order parameter and phase transition has been observed in pump-probe[15,28], near infrared[32,33], and time-resolved ARPES experiments[24,36,43,44], and has unraveled key ingredients in the physics of Ta$_2$NiSe$_5$ in addition to providing a clearer understanding of the nature of the order parameter. The link between the 4.7 THz phonon and the order parameter hints at the possibility that the creation of a photonic time crystal can even be used as a novel method to track order parameter dynamics.

## Results

### Phonon squeezing initiated by photoexcited electrons

**Electron-phonon coupling.** In this section, we develop a microscopic theory of the coupling of electronic bands to IR active phonons. From symmetry arguments, the IR phonon coordinate is odd under inversion, and cannot couple linearly to the electron density which is even under inversion. However, it can couple linearly to electronic dipole transitions which are odd under inversion. For two electronic bands with an allowed dipole transition, this coupling takes the form:

$$H_{el-ph} = Q \sum_k \lambda_k \left( \hat{c}_{1,k}^\dagger \hat{c}_{2,k} + \hat{c}_{2,k}^\dagger \hat{c}_{1,k} \right), \tag{3}$$

where $Q$ is a zero momentum IR phonon coordinate, $\hat{c}_1$ and $\hat{c}_2$ the annihilation operators of the two electronic bands and $\lambda_k$ the electron-phonon coupling matrix element as a function of momentum $k$. Using a Schrieffer-Wolff transformation, we "integrate out" the linear electron-phonon coupling, which is non-resonant due to the different energy scales between the IR-phonon and the electronic transition, to reveal the resonant non-linear coupling between the electron occupation number and the phonon squeezing operator, $Q^2$. In the Methods, we show that this procedure leads to the effective coupling:

$$H_{el-ph,eff.} = \sum_k \frac{\lambda_k^2}{\Delta_k} (n_{1,k} - n_{2,k}) Q^2, \tag{4}$$

where $\Delta_k = E_{1,k} - E_{2,k}$ is the energy difference between the electronic states and $\hat{n}_{i,k} = \hat{c}_{i,k}^\dagger \hat{c}_{i,k}$ the number operator. The above equation applies to phonons that are coupled to independent pairs of electronic bands. If three or more bands are simultaneously coupled to a specific phonon the effective electron-phonon Hamiltonian is more complicated but with similar qualitative features, such as the coupling of electron density to the square of the phonon coordinate.

**Phonon squeezing.** Once electrons have been photoexcited, the finite number of optically excited electrons quenches the frequency of the phonon:

$$H_{el-ph,eff.} = \frac{Mf(t)Q^2}{2}, \tag{5}$$

where $M$ is the mass of the phonon. In equation (5), the parametric driving $f(t)$ comes from the photoexcited distribution of electrons that is strongly coupled to the phonon:

$$f(t) = \sum_k \frac{2\lambda_k^2}{M\Delta_k} \left( \langle n_{1,k} \rangle(t) - \langle n_{2,k}(t) \rangle \right), \tag{6}$$

The photoexcited electron dynamics are fast and can be approximated as a delta function in time. More generally we can approximate the photoexcited distribution as having a characteristic lifetime, $f(t) = f_0 \theta(t) e^{-t/t_{decay}}$, but this does not change our conclusions.

The above electron-induced drive describes a Raman process that does not excite the phonon directly (i.e. $\langle Q \rangle = 0$). However, the squeezing operator starts oscillating at a frequency equal to twice the phonon frequency, as shown in the Methods:

$$\left( \partial_t^2 + \left( 2\omega_{ph} \right)^2 \right) \langle Q^2 \rangle = -2f(t) \langle Q^2 \rangle_0, \tag{7}$$

where $\langle Q^2 \rangle_0$ is the equilibrium fluctuations and $\omega_{ph}$ is phonon frequency at zero momentum which is renormalized by the coulomb force, $\omega_{ph}^2 = \omega_{ph,0}^2 + \frac{Z^2}{\epsilon\epsilon_0 M}$.

To show that this phenomenon is related to squeezing, we expand $Q^2$ in terms of creation and annihilation operators, using $Q = \frac{a + a^\dagger}{\sqrt{2M\omega_{ph}}}$:

$$Q^2(t) = \frac{1}{2M\omega_{ph}}\left(a^\dagger(t)a^\dagger(t) + a(t)a(t) + a^\dagger(t)a(t) + a(t)a^\dagger(t)\right). \quad (8)$$

Expectation values of $a(t)a(t)^\dagger$ do not oscillate rapidly while the anomalous pairs $a^\dagger(t)a^\dagger(t)$ and $a(t)a(t)$ oscillate at twice the phonon frequency. As a result, a state with phonon fluctuations, $Q^2$, that oscillates at twice the phonon frequency implies the existence of a condensate of phonon pairs $\langle a^\dagger(t)a^\dagger(t)\rangle \neq 0$.

**Parametric amplification of reflectivity in pumped Ta$_2$NiSe$_5$**
**Equilibrium reflectivity.** The reflectivity of a material is captured by the frequency-dependent refractive index appearing in the Maxwell equations:

$$\left(\frac{n(\omega)^2\omega^2}{c^2} - k^2\right)E = 0 \quad (9)$$

where $E$ is the electric field and all information about phonons and other IR active modes is encoded in $n(\omega)$. We assume that the probe corresponds to an electromagnetic wave reflected from the sample at normal incidence. The propagation direction is along the $b$-axis of the crystal, which we refer to as the $y$-direction, whereas the electric field points in the $a$-direction which we refer to as the $z$-axis. The refractive index can be directly extracted experimentally from the complex reflection coefficient at normal incidence in an equilibrium system[45]:

$$n(\omega) = \frac{1 - r(\omega)}{1 + r(\omega)}. \quad (10)$$

**Eigenstates in the driven state.** Once we have obtained the refractive index from the equilibrium reflectivity, we model the Floquet material as experiencing a parametric drive oscillating at frequency, $\omega_d$, which mixes signal and idler frequencies. Solutions of coupled light-matter equations are given by Floquet-type eigenmodes[2]:

$$E(t) = e^{iky}\left(E_s e^{-i\omega_s t} + E_{id}e^{i\omega_{id}t}\right). \quad (11)$$

where $\omega_s$ is the frequency of the incoming probe and $\omega_{id} = \omega_d - \omega_s$. Such mixing corresponds to degenerate perturbation theory in Floquet systems and the idler component is the nearest Floquet band contribution to $\omega_s$ which is responsible for parametric instabilities[9,46,47]. The oscillating mode is included phenomenologically through a time-periodic contribution to the electric permittivity:

$$\delta\epsilon(t) = 2A_{\text{drive}}\cos(\omega_d t). \quad (12)$$

Using the ansatz in equation (11), the equations of motion in the Floquet state for the different oscillating components of the electric field become:

$$\left(\frac{n(\omega_s)^2\omega_s^2}{c^2} - k^2\right)E_s + A_{dr}E_{id} = 0 \quad (13a)$$

$$\left(\frac{n(\omega_{id})^2\omega_{id}^2}{c^2} - k^2\right)E_{id} + A_{dr}E_s = 0. \quad (13b)$$

To compute the reflectivity at normal incidence, we first find the allowed $k$ values for a given $\omega_s$. Due to the coupling of signal and idler components, two such $k$ values exist, associated with two transmission channels, both of which oscillate at signal and idler frequencies. The transmission channels correspond to eigenvectors of the Floquet equations of motion inside the material,

$$E_i = t_i E_0 e^{ik_i y}\left(e^{-i\omega_s t} + \alpha_i e^{i\omega_{id}t}\right), \quad (14)$$

where $\alpha_i$ is the relative amplitude of the signal and idler component in the eigenvector of wave-vector $k_i$, $t_i$ is the transmission coefficient of the $i$th channel and $E_0$ is the amplitude of the incoming field.

**Floquet-Fresnel equations.** In a reflection problem, the eigenvalue equation (13) enables the computation of the transmitted wavevectors as a function of a fixed frequency set by the incoming light. However, the answer is given in terms of $k_i^2$ rather than $k_i$, and the correct root is chosen such that the field vanishes at infinity, $\text{Im}\{k_i\} > 0$. The reflectivity is computed by solving the Floquet-Fresnel equations, matching electric and magnetic fields parallel to the surface both at frequency $\omega_s$ and at frequency $\omega_{id}$. Inside the material, we have the electric field:

$$E_{mat} = E_0 \sum_i t_i e^{ik_i y}\left(e^{-i\omega_s t} + \alpha_i e^{i\omega_{id}t}\right), \quad (15)$$

and for vacuum, we have:

$$E_{vac} = E_0\left(e^{i\omega_s/cy - i\omega_s t} + r_s e^{-i\omega_s/cy - i\omega_s t}\right) + E_0 r_{id}e^{i\omega_{id}t + i\omega_{id}y}. \quad (16)$$

Using the homogeneous Maxwell equations, $\nabla \times E = -\partial_t B$ to compute the magnetic field and matching boundary conditions at $y = 0$, we obtain the Fresnel equations for the driven system:

$$1 + r_s = t_1 + t_2, \quad (17a)$$

$$1 - r_s = \frac{k_1}{\omega_s}t_1 + \frac{k_2}{\omega_s}t_2, \quad (17b)$$

$$r_{id} = t_1\alpha_1 + t_2\alpha_2, \quad (17c)$$

$$r_{id} = \frac{k_1}{\omega_{id}}t_1\alpha_1 + \frac{k_2}{\omega_{id}}t_2\alpha_2. \quad (17d)$$

To fit the data, we choose a drive at 9.4 THz motivated by the DFT calculations outlined in the ab initio calculations section. We then use equations (13) and (17) and fit the parameters $A_{drive}$ and $n(\omega)$ to the experimental data allowing for small changes in the static properties of the system such as a photoinduced conductivity stemming from photoexcited charge carriers. The fitted parameters are given in the Methods section.

**Photonic time crystal from squeezing dynamics of phonons.** Material properties such as the electric permittivity, become time-periodic in the presence of oscillating fields through interactions. Here, we demonstrate how lattice potential anharmonicities lead to a time-periodic index of refraction:

We consider a phonon system with a $Q^4$ anharmonicity for the IR-phonon with a Hamiltonian:

$$H_{ph} = \sum_k \left(ZE_k Q_{-k} + M\left(\omega_{ph,0}^2 + f(t)\right)Q_k Q_{-k} + \frac{\Pi_k \Pi_{-k}}{M} + uQ_{k=0}^2 Q_k Q_{-k}\right), \quad (18)$$

where we explore how the squeezing oscillations of the phonon at zero momentum, $Q_{k=0}$, affects the propagation of light at finite $\{E_q, Q_q\}$. The equations of motion for the phonon given by the Hamiltonian in

equation (18) is

$$\left( \partial_t^2 + \gamma \partial_t + \omega_{ph,0}^2 + f(t) + u Q_{q=0}^2 \right) Q_k = \frac{Z}{M} E_k. \tag{19}$$

To describe the dynamics of phonons, we include phenomenologically the damping term $\gamma$, which captures the decay of the phonon to other degrees of freedom. Using a Gaussian ansatz for the phonons, we can linearize the above equation as:

$$\left( \partial_t^2 + \gamma \partial_t + \omega_{ph,0}^2 + u \langle Q_{q=0}^2 \rangle(t) \right) Q_k = \frac{Z}{M} E_k, \tag{20}$$

where the fluctuations $\langle Q^2 \rangle = \langle Q^2 \rangle_0 + A \cos(2\omega_{ph} t)$. The contribution from $f(t)$ corresponds to the fast dynamics of the photoelectrons and therefore has a broad spectrum. As such, it does not lead to resonant dynamics and will at most contribute to an incoherent background. In contrast, squeezing oscillations oscillating at a specific frequency can lead to parametric amplification that is narrowband. Hence, we drop the contribution of $f(t)$ and focus on the consequences of the squeezing mode oscillations. The phonon mode appears in the Maxwell equations as:

$$\left( \frac{1}{c^2} \partial_t^2 - k^2 \right) E_k = -Z \partial_t^2 Q_k. \tag{21}$$

To find the effective signal idler mixing presented in equation (13), we expand the equations of motion in signal and idler contributions:

$$Q_k = Q_s e^{-i\omega_s t} + Q_{id} e^{i\omega_{id} t} \tag{22}$$

Equation (20) becomes:

$$\begin{pmatrix} Q_s \\ Q_{id} \end{pmatrix} = \begin{pmatrix} \frac{Z/M}{\omega_s^2 + i\gamma\omega_s - \omega_{ph}^2} & 0 \\ 0 & \frac{Z/M}{\omega_{id}^2 + i\gamma\omega_{id} - \omega_{ph}^2} \end{pmatrix} \cdot \begin{pmatrix} E_s \\ E_{id} \end{pmatrix} - \frac{Z/MuA}{2\left(\omega_s^2 + i\gamma\omega_s - \omega_{ph}^2\right)\left(\omega_{id}^2 + i\gamma\omega_{id} - \omega_{ph}^2\right)} \begin{pmatrix} E_{id} \\ E_s \end{pmatrix}. \tag{23}$$

Substituting equation (23) in Maxwells equation we find the equations of motion for the signal and idler component of the electric field to be:

$$\left( \frac{n_{eq.}^2(\omega_s)}{c^2} \omega_s^2 - k^2 \right) E_s + A_{drive,s}(\omega_s, \omega_{id}) E_{id} = 0, \tag{24a}$$

$$\left( \frac{n_{eq.}^2(\omega_{id})}{c^2} \omega_s^2 - k^2 \right) E_s + A_{drive,id}(\omega_s, \omega_{id}) E_{id} = 0 \tag{24b}$$

where the signal and idler driving amplitudes, $A_{drive,s}$ and $A_{drive,id}$ are given by:

$$A_{drive,s} = \frac{-Z^2/MuA\omega_s^2}{2\left(\omega_s^2 + i\gamma\omega_s - \omega_{ph}^2\right)\left(\omega_{id}^2 + i\gamma\omega_{id} - \omega_{ph}^2\right)}, \tag{25a}$$

$$A_{drive,id} = \frac{-Z^2/MuA\omega_{id}^2}{2\left(\omega_s^2 + i\gamma\omega_s - \omega_{ph}^2\right)\left(\omega_{id}^2 + i\gamma\omega_{id} - \omega_{ph}^2\right)}. \tag{25b}$$

The above analysis presents an illustration of how non-linearities and squeezing dynamics lead to a Photonic time crystal. In this specific model, we find that the signal idler coupling is proportional to the IR activity of the mode, $Z$, the phonon anharmonicity, $u$, and the

amplitude of the oscillations, $A$: $A_{drive} \propto Z^2 uA$. The precise strength of the coupling between signal and idler modes will depend on the choice of non-linearity. As a result, in this article, we take an agnostic view and instead fit the experimental data to the signal and idler coupling in equation (13).

## Ab initio calculations: the 4.7 THz IR-Phonon

The above discussion is generic and in principle applies to every IR phonon inside a given material. In this section, we use DFT calculations to determine which phonons make the dominant contribution to the parametric amplification of reflectivity in T$a_2$NiSe$_5$. We start our discussion by identifying which valence and conduction bands are involved in photo-absorption by evaluating the optical dipole transition matrix elements. In Fig. 2(a), we plot the momentum-resolved optical contribution, for field polarization along the $a$-axis, of each band to dipole-allowed transitions within the experimentally relevant energy window between 0.33 eV and 0.9 eV set by the pump parameters. For a given valence(conduction) band and k-point, the optical contribution is defined by summing the square of the dipole transition matrix elements associated with the transition. We find that it is the first three conduction bands that are predominantly excited by the pump.

Turning our attention to phonons, we use ab initio calculations to identify all IR active phonons. In particular, we find a number of phonons in the low-temperature monoclinic phase between 4 and 5 THz presented in Supplementary Table 1 in Supplementary Note 1. In Fig. 2 (b) we plot the IR activity of phonons as a function of frequency and identify the 4.7 THz phonon which, as shown below, turns out to be the most strongly electron-coupled phonon.

We compute the electron-phonon interaction between IR active phonons and electrons. This is to identify the phonons that dominate the interaction with the photo-excited electronic bands. To accomplish this we use the method of frozen phonons. In this approach, the electronic bands are recalculated with the lattice shifted along a phonon eigendisplacement. To quantify the coupling strength of a specific phonon to a particular electronic band we integrate the energy changes of the band in the presence of the phonon over the Brillouin zone as outlined in Supplementary Note 1. We find that in the vicinity of 4.5 THz, relevant to the experimental observables, the 4.7 THz mode has roughly an order of magnitude larger electron-phonon coupling strength compared to the nearby phonons. In particular, as shown in Fig. 2c, the 2nd and 3rd conduction bands (in this paper we number the conduction bands from lowest to highest in energy) are significantly renormalised in the presence of the 4.7 THz phonon. Since the phonon is mostly coupled to two electronic bands, the frozen phonon calculations are consistent with equation (4) leading to the two bands shifting in energy by an equal and opposite amount given by:

$$E_{k,3}|_{\langle Q \rangle \neq 0} - E_{k,3}|_{\langle Q \rangle = 0} = -\left( E_{k,2}|_{\langle Q \rangle \neq 0} - E_{k,2}|_{\langle Q \rangle = 0} \right)$$

$$= \langle Q \rangle^2 \frac{\lambda_k^2}{\Delta_k \left( 1 - \frac{\omega_{ph,0}^2}{\Delta_k^2} \right)} \tag{26}$$

where $E_{k,2}$ and $E_{k,3}$ is energy of conduction bands two and three at momentum $k$, and $\Delta_k$ is the energy difference of the two bands in equilibrium. Therefore, in this case, calculating the energy shifts as a function of momentum in the frozen phonon approximation allows for direct computation of the electron-phonon interaction.

To summarize, we use ab initio calculations to identify the electron bands excited by pumping and to establish the IR-active phonons in T$a_2$NiSe$_5$. Subsequently, frozen phonon calculations allowed for the identification of the phonon with the dominant electron-phonon coupling to the excited electronic bands. This leads to the identification of the 4.7 THz mode as responsible for the parametric

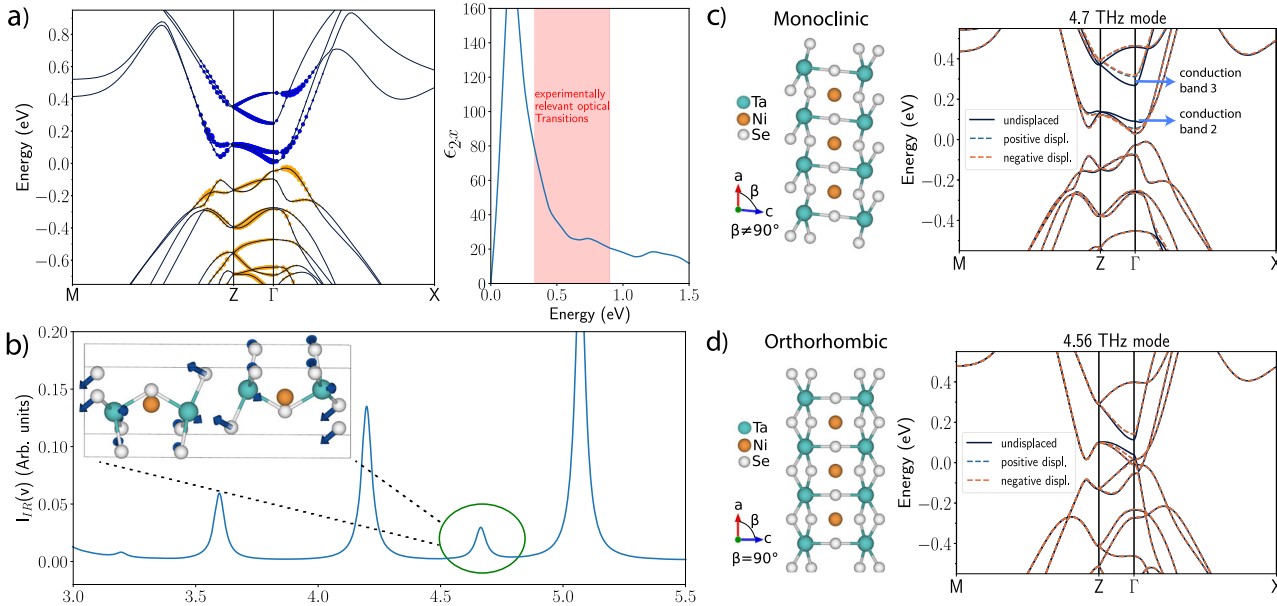

**Fig. 2 | Summary of ab initio calculations. a** Left-hand side panel: Optical matrix elements along the *a*-axis calculated for the frequency region between 0.4 to 0.9 eV. The size of the circles is proportional to the magnitude of the matrix elements and indicates which electronic valence(yellow) and conduction(blue) bands are involved in photo-absorption. Right-hand side panel: Imaginary part of permittivity along the *a*-axis as a function of frequency with the experimentally relevant frequency window shaded. **b** IR activity of the most IR active phonons; the phonon at 4.7 THz which dominates electron-phonon interactions in that region is highlighted with the mode character shown in the inset. A complete list of IR active phonons in

this region for TNS is given in Methods. **c** and **d** show the effect of electron-phonon coupling on the band structure in the monoclinic and orthorhombic phases respectively. The electron-phonon coupling is captured by calculating the electronic band-structures using DFT in the presence of either negative or positive phonon displacement (frozen phonon approximation). The phonon 4.7 THz in the monoclinic phase is strongly coupled to the electronic bands 2 and 3 which are highlighted in the figure. In the high-temperature orthorhombic phase, the effective electron-phonon coupling is reduced by an order of magnitude. This indicates that the electron-phonon interaction is very sensitive to the order parameter.

amplification observed in experiments through the mechanism outlined in previous subsections. We note that we do not exclude the possibility of subdominant amplification in reflectivity spectra arising from other phonons at different frequencies. In Supplementary Note 1, we provide more details of our DFT analysis of IR-phonons in T$a_2NiSe_5$, the frozen phonon calculations, and the calculation of optical matrix elements.

Therefore, the DFT calculation provides strong support for the minimal electron-phonon model and uncovers the dominant electronic bands and IR phonons involved in the nonlinear process. The other ingredients needed to extract the amplification amplitude, i.e., phonon nonlinearities, phonon dissipation, and the dynamics of the photoexcited electrons responsible for phonon squeezing, were included phenomenologically as described in the photoinduced reflectivity fitting section.

**Connection to the order parameter.** Finally, we discuss the effects of the phase transition on the phonon-squeezing process. In Fig. 2, we compute the electron-phonon interaction of the 4.7 THz phonon in the low temperature monoclinic phase while for the high temperature orthorhombic phase we compute the electron-phonon interaction of the phonon adiabatically connected to the 4.7 THz phonon eigenstate (for details on the identification of the adiabatically connected phonon see Supplementary Note 1 and Supplementary Fig. 1). The electron-phonon interaction effectively disappears in the high-temperature phase providing evidence that electron-phonon coupling is mediated by the order parameter. In particular, we suggest that close to the phase transition the electron-phonon coupling is proportional to the order parameter, $\lambda_k = \Phi B_k$, where $\Phi$ is the order parameter and $B_k$ a constant. This, in turn, suggests that parametric amplification can be used as a nontrivial probe to investigate order parameter dynamics. The microscopic reason for the strong dependence of the electron-

phonon coupling to the order parameter is a very interesting question that could reveal new insights about the nature of the order parameter in T$a_2NiSe_5$ and will be addressed in a subsequent publication.

## Discussion

We have investigated the microscopic mechanism of amplification of THz optical reflectivity in $Ta_2NiSe_5$ arising from high-frequency optical pumping. We showed that strong electron-phonon coupling opens new pathways toward realizing THz parametric amplification through high-frequency pumping. Ab initio calculations highlight the importance of the 4.7 THz IR-active phonon which is strongly coupled to electrons allowing for the amplification to manifest in the THz reflection spectrum. This parametric process is attributed to the phonon squeezing phenomenon which transiently drives the material into a photonic time crystal.

Our theory indicates that choosing which electronic band to photoexcite selects the IR phonon that is most strongly coupled to that electronic band. As a result, we can use different pumping frequencies in the same material to tune the frequency of the THz parametric amplification through mode-selective phonon squeezing.

Finally, we showed that the electron-phonon coupling is strongly dependent on the order parameter and becomes suppressed in the high-temperature orthorhombic phase. This suggests that time crystalline behaviours in $Ta_2NiSe_5$ may be used as a new probe to order parameter dynamics.

## Methods

### The theoretical fit of the parametric reflectivity amplification in pumped T$a_2NiSe_5$

As mentioned in the main text in equation (10), in principle the complex refractive index can be directly computed by the complex reflectivity amplitude, $r(\omega)$. However, small phase errors upon experimental

extraction of $r(\omega)$ could lead to the unphysical behaviour of the reflectivity. To overcome this complication, we fit the data by assuming that for an insulator like $Ta_2NiSe_5$ the refractive index is real. As a result, we can instead use the absolute value of $|r| = \sqrt{R}$ and express the refractive index as:

$$n(\omega) = \frac{1 + \sqrt{R}}{1 - \sqrt{R}}. \tag{27}$$

Upon parametric resonance and solving equation (13) together with the Floquet-Fresnel boundary conditions in equation (17), assuming a purely real spectrum with no dissipation leads to highly divergent behavior on parametric resonance. As a result, we put back dissipation by including an imaginary component to the refractive index. In the driven case we argue that this is physical since $n_{\text{driven}}(\omega)^2 = n^2(\omega) + i\sigma_{\text{driven}}/\omega$ and dissipation can arise through a transient contribution to the conductivity by electrons excited across the gap of the insulator. To fit the data we choose a parametric drive at $\omega_d = 9.4$ THz, drive amplitude $A_{\text{drive}} = 7.5 \frac{THz^2}{c^2}$, overall constant renormalization of the refractive, $n_{\text{drive}}^2 = n^2(\omega) + \delta n^2$, with $\delta n^2 = 0.1 + 0.01i$ and an overall Gaussian broadening function with a standard deviation of 0.1 THz. In particular, $\omega_d = 9.4$ THz was set to match the squeezing fluctuations of the 4.7 THz mode, which we derived from DFT as being the most strongly coupled mode to the electrons in that frequency region. The fitted $A_{\text{drive}}$ corresponds to a small time-dependent change in permittivity compared to the equilibrium permittivity, $\frac{A_{\text{drive}}}{\omega^2 n^2(\omega)/c^2} \sim 0.01 - 0.1$.

We note in the present work as mentioned earlier in this section, it is not possible to experimentally estimate the imaginary part of the refractive index, accounting for dissipation. This leads to an ambiguity in $A_{\text{drive}}$, since the presence of more dissipation would need a larger drive amplitude to achieve the same amplification. We found the best fit by fitting both $A_{drive}$ and dissipation together. Careful determination of the equilibrium complex reflectivity would allow us to better constrain this parameter from the photoinduced reflectivity.

## Experimental details

For the experimental data, a state-of-the-art reflection-based optical pump - THz probe spectroscopy system was used[23]. In the pump-probe setup, the sample was excited with 45-fs 0.5 eV (2.4 μm) pulses generated from a Ti: sapphire laser system having 1 KHz repetition rate, 3 mJ pulse energy, and 800 nm central frequency, via a Light Conversion TOPAS-C optical parametric amplifier. Broadband THz probe beams were generated by a 2-color air-based laser plasma scheme[48,49].

For the detection, the electro-optic sampling method was employed with a GaP sampling crystal which enabled spectral access to a broad energy window of 0.4 to 7.5 THz. A 1 MHz bandwidth Newport 2307 balanced photodetector was utilized to detect the signal. For the acquisition of the data, a NI PXIe-5122 data acquisition system (DAQ) with a 100 MHz maximum sampling rate and 100 MHz bandwidth was used.

Subsequently, the THz reflectivity of the $Ta_2NiSe_5$ was obtained using a gold mirror as the reference. We measured $E_{gold}(t)$ and $E_{sample}(t)$, which are the reflections of the incident THz probe field from the reference and sample, respectively. Here, $t$ is the electro-optic gate time. Fast-Fourier transforming the time-domain signals yield frequency domain responses $E_{gold}(\omega)$ and $E_{sample}(\omega)$. The equilibrium reflectivity was computed as $R = |E_{sample}(\omega)/E_{gold}(\omega)|^2$. The pump-induced transient change in the reflected probe electric field was measured as $\Delta E(t)$ which can be written as $\Delta E(t) = E_{sample,pumped}(t) - E_{sample,unpumped}(t)$. Using the pump-probe signal $\Delta E(t)$, the photoinduced reflectivity $R + \Delta R$ can be calculated as $R + \Delta R = |(E_{sample}(\omega) + \Delta E(\omega))/E_{gold}(\omega)|^2$.

Further, pump-induced change in reflectivity $\Delta R$ was computed by subtracting the equilibrium reflectivity spectrum $R$ (pump off) from photoinduced reflectivity spectrum $R + \Delta R$ (pump on). As such, the $\frac{\Delta R}{R}$ spectrum as shown in Fig. 1(c) essentially denotes the enhancement in THz reflectivity upon photoexcitation.

## Electron-phonon interaction

The Hamiltonian of two electronic bands coupled by an allowed direct dipole transition that is, in turn, coupled to a phonon is given by:

$$H_{el} = \frac{\Delta}{2}\left(\hat{c}_1^\dagger \hat{c}_1 - \hat{c}_2^\dagger \hat{c}_2\right) + \lambda\left(\hat{c}_1^\dagger \hat{c}_2 + \hat{c}_2^\dagger \hat{c}_1\right)Q, \tag{28}$$

where $\Delta = E_1 - E_2$ is the difference in energy between the two bands, $\lambda$ is the coupling constant coming from the dipole-dipole interaction in the dipole gauge, $Q$ is the phonon coordinate and $\{\hat{c}_1, \hat{c}_2\}$ are the annihilation operators of the two-electron bands. The IR-phonon quadratic Hamiltonian is given by:

$$H_{ph} = ZEQ + M\omega_{ph,0}^2 \frac{Q^2}{2} + \frac{\Pi^2}{2M}, \tag{29}$$

where $\Pi$ is the conjugate momentum of the phonon coordinate $Q$, $Z$ is the effective coupling to the electromagnetic field, $E$, and $\omega_{ph,0}$ is the phonon frequency. Since the transition itself is not resonant with the phonon mode, we decouple the linear electron-phonon interaction perturbatively using a Schrieffer-Wolff transformation. This generates an interaction between the electron-hole pair density and the fluctuations of the phonon field which can be a resonant process. To perform this transformation, it is convenient to note that bilinear combinations of $\{\hat{c}_1^\dagger, \hat{c}_1, \hat{c}_2^\dagger, \hat{c}_2\}$ appearing in the Hamiltonian, obey SU(2) commutation relations by making the following identification:

$$S^x = \frac{\hat{c}_1^\dagger \hat{c}_2 + \hat{c}_2^\dagger \hat{c}_1}{2}, \tag{30a}$$

$$S^y = -i\frac{\hat{c}_1^\dagger \hat{c}_2 - \hat{c}_2^\dagger \hat{c}_1}{2}, \tag{30b}$$

$$S^z = \frac{\hat{c}_1^\dagger \hat{c}_1 - \hat{c}_2^\dagger \hat{c}_2}{2}, \tag{30c}$$

where their commutators are $[S^x, S^y] = iS^z$ and their cyclic permutations. In terms of the spin operators, we separate the Hamiltonian into a non-interacting and an interacting part:

$$H_0 = \Delta S^z + M\omega_{ph,0}^2 \frac{Q^2}{2} + \frac{\Pi^2}{2M}, \tag{31}$$

$$V = 2\lambda S^x Q \tag{32}$$

To remove the interacting part $V$ to linear order in $\lambda$, we consider a unitary transformation of the type:

$$U = \text{Exp}\{iA\}, \tag{33}$$

$$A = \alpha Q S^y + \beta \Pi S^x \tag{34}$$

The Schrieffer-Wolff expansion is given by:

$$U H_{total} U^\dagger = H_0 + V - i[H_0, A] - i[V, A] - \frac{1}{2}\left[[H_0, A], A\right]. \tag{35}$$

The parameters $\alpha$ and $\beta$ are found such that:

$$V = i[H_0, A] \tag{36}$$

Matching linear terms in $\Pi$ and $Q$, leads to the parameters:

$$\beta = \frac{\alpha}{M\Delta}, \tag{37a}$$

$$-\Delta\alpha + 2\lambda + M\omega_{\text{ph},0}^2\beta = 0, \tag{37b}$$

$$\Rightarrow \alpha = \frac{2\lambda}{\Delta\left(1 - \frac{\omega_{\text{ph},0}^2}{\Delta^2}\right)}, \tag{37c}$$

which confirms that this perturbation theory can be carried out as long as $\omega_{\text{ph},0}$ is off-resonant with the transition energy $\Delta$. The effective electron-phonon interaction after the Schrieffer-Wolff transformation is given by:

$$\begin{aligned} H_{\text{eff}} &= -i[V, A] - \frac{1}{2}\big[[H_0, A], A\big] = -\frac{i}{2}[V, A], \\ &= \frac{2\lambda^2}{\Delta\left(1 - \frac{\omega_{\text{ph},0}^2}{\Delta^2}\right)}Q^2 S^z + \frac{2\lambda^2}{\Delta\left(1 - \frac{\omega_{\text{ph},0}^2}{\Delta^2}\right)}(S^x)^2, \end{aligned} \tag{38}$$

where the second term does not depend on the phonons. The residual electron-phonon interaction is thus given by:

$$H_{\text{el-ph,eff}} = \frac{\lambda^2}{\Delta\left(1 - \frac{\omega_{\text{ph},0}^2}{\Delta^2}\right)}Q^2(n_1 - n_2) \tag{39}$$

where $n_1 = \hat{c}_1^\dagger \hat{c}_1$ is the occupation number of mode 1. This implies that if either mode 1 or mode 2 are photoexcited the phonon can be squeezed.

The linear term in the electric field, $ZEQ$, appearing in equation (29), is also transformed by the Schrieffer-Wolff transformation. It gives rise to a term $\beta ZES^x$ which provides a small renormalization of the dipole transition amplitude between the bands and does not affect our discussion.

The above result can be generalized to an arbitrary number of independent pairs of electronic states. For example, for two electronic states, the above result in the limit is generalized to:

$$H_{\text{el-ph,eff}} = \sum_k \frac{\lambda_k^2}{\Delta_k}Q^2(n_{1,k} - n_{2,k}), \tag{40}$$

This expression is the one used in the main text.

**Phonon squeezing**

Using a Hartree-Fock type approximation on the effective electron-phonon Hamiltonian in equation (40), we derive an effective Hamiltonian for the phonon system only:

$$H_{\text{ph}} = ZEQ + M\left(\omega_{\text{ph},0}^2 + f(t)\right)\frac{Q^2}{2} + \frac{\Pi^2}{2M} \tag{41}$$

where $Z$ is the IR activity of the phonon mode, $\omega_{\text{ph},0}$ the bare phonon frequency and the effective parametric drive, $f(t)$, is given by photoexcited electron density coupled to the phonon mode:

$$f(t) = \sum_k \frac{2\lambda_k^2}{M\Delta_k}\left(\langle n_{1,k}\rangle - \langle n_{2,k}\rangle\right), \tag{42}$$

Due to the fast dynamics of electrons, $f(t)$ acts as an impulsive delta function like a parametric drive. Such a drive is not periodic but it can linearly excite phonon fluctuations $\langle Q^2\rangle$ which will oscillate in time. To show this, we compute the equations of motion for fluctuations of the phonon field:

$$\partial_t\langle Q^2\rangle = \frac{\langle \Pi Q + Q\Pi\rangle}{M}, \tag{43a}$$

$$\partial_t\langle \Pi Q + Q\Pi\rangle = -2M\left(\omega_{\text{ph},0}^2 + f(t)\right)\langle Q^2\rangle + 2\frac{\langle\Pi^2\rangle}{M} - 2Z\langle EQ\rangle, \tag{43b}$$

$$\partial_t\langle\Pi^2\rangle = -M\left(\omega_{\text{ph},0}^2 + f(t)\right)\langle(\Pi Q + Q\Pi)\rangle - Z\langle E\Pi + \Pi E\rangle. \tag{43c}$$

At $k = 0$, we can use Maxwell's equations to remove the electric field dependence, $E = \frac{Z}{\epsilon\epsilon_0}Q$. Performing this substitution simplifies the equations of motion,

$$\partial_t\langle Q^2\rangle = \frac{\langle\Pi Q + Q\Pi\rangle}{M}, \tag{44a}$$

$$\partial_t\langle\Pi Q + Q\Pi\rangle = -2M\left(\omega_{\text{ph}}^2 + f(t)\right)\langle Q^2\rangle + 2\frac{\langle\Pi^2\rangle}{M}, \tag{44b}$$

$$\partial_t\langle\Pi^2\rangle = -M\left(\omega_{\text{ph}}^2 + f(t)\right)\langle(\Pi Q + Q\Pi)\rangle. \tag{44c}$$

where $\omega_{\text{ph}}^2 = \omega_{\text{ph},0}^2 + \omega_{\text{pl,phonon}}^2$ is the frequency of the phonon at $k = 0$ which differs from the bare frequency by the phonon plasma frequency given by $\omega_{\text{pl,phonon}}^2 = \frac{Z^2}{\epsilon\epsilon_0 M}$. Being perturbative in the drive $f(t)$, we expand the phonon fluctuations,

$$\langle Q^2\rangle = \langle Q^2\rangle_0 + \langle Q^2\rangle_1, \tag{45}$$

where $\langle Q^2\rangle_0$ is the thermal expectation value and $\langle Q^2\rangle_1 \propto f(t)$. To linear order in $f(t)$, the equations of motion imply, $\langle\Pi^2\rangle_1/M = -M\omega_{\text{ph}}^2\langle Q^2\rangle_1 + \mathcal{O}(f^2)$. Finally, combining equations in (44) we find that:

$$\left(\partial_t^2 + 4\omega_{\text{ph}}^2\right)\langle Q^2\rangle_1 = -2f(t)\langle Q^2\rangle_0. \tag{46}$$

This result shows that phonon fluctuations are linearly driven by photoexcitation and oscillate at twice the phonon frequency, $2\omega_{\text{ph}}$. These coherent oscillations of phonon fluctuations behave as a Raman mode.

In our treatment, we implicitly assume that only $k = 0$ phonons are excited by photoelectrons. The contribution of finite momentum squeezing modes is beyond the scope of our work.

## Data availability
All the data and supporting materials can be made available from the corresponding authors upon request. The source data for Fig. 1c, which represents the main result of the current paper, are provided in this paper. Correspondence should be addressed to Marios H. Michael or Sheikh Rubaiat Ul Haque. Source data are provided in this paper.

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

## Acknowledgements

We acknowledge the fruitful discussion with Jonathan B. Curtis, Mohammad Hafezi, Andrey Grankin, Daniele Guerci, John Sous, Andy Millis, Martin Eckstein, and Hope Bretscher. M.H.M. is grateful for the financial support received from the Alexander von Humboldt post-doctoral fellowship. M.H.M., S.R.U.H., Y.Z., E.D., and R.D.A. acknowledge support from DARPA 'Driven Nonequilibrium Quantum Systems' (DRINQS) program under award no. D18AC00014. E.D. also acknowledges support from the ARO grant "Control of Many-Body States Using Strong Coherent Light-Matter Coupling in Terahertz Cavities" and the SNSF project 200021_212899. L.W., S.L. and A.R. acknowledge support from the European Research Council (ERC-2015-AdG694097), the Cluster of Excellence 'Advanced Imaging of Matter' (AIM), Grupos

Consolidados (IT1249-19) and Deutsche Forschungsgemeinschaft (DFG) - SFB-925 - project 170620586. A.R. also thanks the Flatiron Institute, a division of the Simons Foundation.

## Author contributions

E.D. and R.D.A. conceived the project together with M.H.M. and S.R.U.H. M.H.M. and E.D. developed the theory framework, analyzed the data, and performed numerical simulations. The optical pump - THz probe data was provided by S.R.U.H. and Y.Z. L.W., S.L., and A.R. performed the DFT analysis. All authors participated in the discussion and interpretations of the results. E.D. and R.D.A. supervised the project. M.H.M. and S.R.U. H. wrote the manuscript with input from all authors.

## Funding

## Competing interests

The authors declare no competing interests.
