## [Peer Review File · Nature Communications]

REVIEWER COMMENTS

Reviewer #1 (Remarks to the Author):

The manuscript "Theory of time-crystalline behaviour mediated by phonon squeezing in Ta₂NiSe₅" reports a theoretical study of the excitonic insulator candidate Ta₂NiSe₅ and photonic time-crystalline behavior there. The latter is triggered by optical excitation above the electronic gap. The authors demonstrate that following electron photoexcitation, electron-phonon coupling results in an atypical squeezed phonon state that is characterized by phonon fluctuations oscillating periodically. Compressing vibrations achieve crystalline behavior in photonic time. THz amplification of reflectivity in a specific frequency range is argued to be the primary indicator of photonic time-crystalline behavior.

Experimental results on Ta₂NiSe₅ suggest that photoexcitation with short pulses produces enhanced terahertz reflectivity with the predicted characteristics. The developed theory explains the observations reported in arXiv:2304.09249 by almost the same group of authors. The authors describe the fundamental process that results in THz amplification in terms of a reduced Hamiltonian, the validity of which is confirmed by ab initio DFT computations. For the sake of clarity for readers, it would be desirable to merge these two papers into one manuscript and publish it as Nature Communication.

The results reported in the manuscript are interesting and might potentially need to be communicated to the wider specialized community, such as the readership of Nature Communications. However, upon reading the manuscript, we stumbled upon several places that prevented us from appreciating the full picture. We list these questions and suggested improvements below. Without these improvements, we will not be able to recommend the publication of the manuscript.

1. Crystalline substances normally have an orderly and repetitive atomic arrangement. This notion suggests that time-crystalline behavior is similar to time crystals, where, under time translations, the period increases compared to the period of the Hamiltonian. This is exactly what is reported in the present manuscript, which makes us wonder if the suggested terminology is the best. We ask the authors to elaborate on this choice of terminology in the manuscript. In any case, it would help avoid associating new terms with the observed effect and explain its physics in simpler terms accessible to a wider community.

2. The origin of eq 12 is unclear for a non-specialized reader. After screening Ref [48] of the manuscript, it is still somewhat obscure why the linear in Q coupling of the effective dipole of the phonon to the electronic dipole should have the form of Eq 12. We suggest the authors introduce more arguments and discussions supporting this particular coupling. Particularly, it would help to see the derivation of eq 12 based on a general symmetry-based scenario.

3. The theory line in Figure 1c is one of the main messages of the work. We wanted to trace the details and specifics of the calculation, particularly and noticed that it was given by Fresnel eqs. 11. The matching with experiments was achieved upon fixing four parameters (as far as we could trace). The authors should comment on

a) How reasonable are the fixed values of fitting parameters?

b) How much the result will deviate upon changing the values of the fitting parameters.

4. The authors described two phenomena: the 4.7 THz reflection peak under the drive frequency of 9.4 THz (photonic time crystal), and the parametric phonon oscillation at 4.7 THz, whose variance oscillates at 7.4 THz. It is unclear to us how these two are related. It would be helpful if the authors added more discussion on this.

5. A related question to point 4. In fig. 2b, there exist peaks other than the 4.7 THz. Why those peaks don't result in peaks in the reflectivity? Other ab initio DFT parameters and results should be compared to the Fresnel equations. Otherwise, the conclusion "the 4.7 THz mode is responsible for the parametric amplification observed in experiments" is not supported by the data.

Reviewer #2 (Remarks to the Author):

Reviewer #3 (Remarks to the Author):

The present work deals with the amplification of THz reflectivity of an insulating material following electron photoexcitation above the gap. Experimental results are presented and a theoretical model developed, which can be used to well fit the data, and offers a physical explanation of the phenomenon. The latter entails that the photoexcited electron population induces oscillations of the phonon quadrature in a squeezed state, which in turn works as a parametric amplifier of a THz probe tuned around that phonon mode.

My opinion as a non-expert in the field of optically pumped materials is that the present observations are very interesting as well as technologically promising. The physical mechanism identified here seems plausible and generic enough to impact and spur investigations in other materials.

I am therefore in favour of publication. However, I suggest substantial reorganisation of the presentation, which I would break down in the following points:

1. Title. The classification of the behaviour as a "time crystal" seems not very appropriate and also unnecessary for the whole message. In my view, time crystals are interesting because the spontaneous breaking of TTI and the resulting spontaneous oscillations are infinitely long-lived or at least exceptionally robust despite the fact that one is dealing with a macroscopic number of interacting degrees of freedom, among which the energy can in principle be redistributed -- leading to heating and time-crystal melting. In the present work there is no study of the oscillations-lifetime and no argument sustaining their exceptional robustness. Therefore, I suggest to modify the title to "Theory of THz amplification mediated...". In my opinion, pitching the work like this would be more appealing and relevant for the science of quantum materials than referring to the somewhat "synthetic" albeit fundamentally exciting phenomenon of time crystals.

2. Experimental part. I understand that the experimental data is not published anywhere and thus the present manuscript should be regarded as a theory-experiment collaboration. If this is the case, then I find that too little relevance is given to the experimental part, which provides the starting point and orientation for the theory part. The title is clearly classifying this work as a theory work, and in the main text the description of the experimental methods is maybe 5% of the whole, without even a dedicated section.

3. Explanation of the theory. I find the explanation of the model and calculations not easy to follow. This is probably in part caused by my limited expertise in the field, but I find that in a broad-audience journal the basic logic, ingredients, and assumptions should become clear to all readers, and from the main text. Therefore, I suggest some reorganisation. The main suggestion is to present the

section "Phonon-squeezing initiated ... electrons" BEFORE the section "Parametric amplification ... Ta₂NiSe₅". The reason is that I understood the parametric-amplification model (signal-idler mixing) only after I read the Phonon-squeezing section, and in particular the derivation in the METHODS. In particular, I think that few equations from the METHODS-section "Floquet matter from...phonons" need to be moved to the main text, e.g. the full Hamiltonian (39), and eq.(40) or its linearised version, otherwise it is really not clear how the interplay between E-field and phonons works and is modelled. Starting from (39) one could also derive, at the same time, equation (40), describing the coupling of $\langle Q \rangle$ to the THz probe-field and to the quadrature $\langle Q^2 \rangle$ through anharmonicity, as well as equation (38), describing the excitation of the quadrature oscillations from the electron population. This would result in a much more organised and compact presentation, the crucial steps of which can be put in the main text as suggested above.

4. Extent of the ab-initio support. The abstract might sound like the whole theory model is supported by ab-initio calculations. By reading the corresponding section it is clear that the ab-initio part nicely justifies the electron-phonon coupling Hamiltonian, but not the whole subsequent calculations down to the final prediction about amplification. While the latter seem compelling and sound to me, it might be desirable to state more clearly whether and which important assumptions/simplifications are made which are not justified from the ab-initio and would in principle need confirmation through e.g. some kind of ab-initio dynamical simulations.

5. Fitting. It was not clear to be which parameters are fitted and which obtained otherwise. Even if briefly, this seems important to be mentioned in the main text, and reiterated in the detailed derivation of the methods section.

Further more detailed remarks:

a. Eq.(2). This equation seems a bit disconnected from the formalism developed later in the text.

b. Treatment of the E-field. The E-field enters both eq.(35) and eq.(40). However, it is treated differently between the two, suggesting that two different sets of modes are relevant for these equations. It seems that in (35) only the $k=0$ mode is considered (not excited by the probe), while in (40) the signal-idler modes at finite k are considered. Some further discussion would help.

c. From (39) to (40) some damping γ appears without discussion it seems. Please elaborate.

d. In eqs.(40,41) the driving function f has disappeared. Why? I guess it comes from the linearisation as done in the eqs. for $\langle Q^2 \rangle$, but then f should still be there in (40). Please explain.

REVIEWER 1

The manuscript "Theory of time-crystalline behaviour mediated by phonon squeezing in Ta₂NiSe₅" reports a theoretical study of the excitonic insulator candidate Ta₂NiSe₅ and photonic time-crystalline behavior there. The latter is triggered by optical excitation above the electronic gap. The authors demonstrate that following electron photoexcitation, electron-phonon coupling results in an atypical squeezed phonon state that is characterized by phonon fluctuations oscillating periodically. Compressing vibrations achieve crystalline behavior in photonic time. THz amplification of reflectivity in a specific frequency range is argued to be the primary indicator of photonic time-crystalline behavior.

Experimental results on Ta₂NiSe₅ suggest that photoexcitation with short pulses produces enhanced terahertz reflectivity with the predicted characteristics. The developed theory explains the observations reported in arXiv:2304.09249 by almost the same group of authors. The authors describe the fundamental process that results in THz amplification in terms of a reduced Hamiltonian, the validity of which is confirmed by ab initio DFT computations. For the sake of clarity for readers, it would be desirable to merge these two papers into one manuscript and publish it as Nature Communication.

Our response: We thank the referee for their time and effort carefully reviewing our work. We made the decision to present the theory and experiment in separate papers in order to highlight orthogonal messages and implications.

The experimental paper which accompanies our work, has now been published in *Nature Materials* (DOI: <https://doi.org/10.1038/s41563-023-01755-2>, ref. [13] in the current manuscript). In the experimental paper, we highlight how nonlinear processes such as electron-phonon mediated amplification of reflectivity can be used as a reporter for the excitonic order parameter in Ta₂NiSe₅, by focusing on the anomalous temperature dependence of the result. The theory of the amplification there is presented on a surface level.

Conversely, in the current paper, our primary focus is on advancing the theoretical framework, elucidating how a robust electron-phonon coupling can introduce a novel mechanism for generating low-frequency THz photonic time crystals by high-frequency optical pumping. From this perspective, the theory uncovers a universal phenomenon, with experiments on Ta₂NiSe₅ providing a compelling illustration of the effect due to its pronounced electron-

phonon coupling. We believe that a consolidated paper would risk diluting these distinct messages and would end up being too long. We hope that this explanation convinces the referee of the rationale behind our decision to present these aspects separately.

Changes made to the manuscript:

We appreciate the Referee’s suggestion that the two papers need to be combined for the sake of clarity for the readers. To address this, we have now added an extended Experimental details section in the Methods, which we now point to in the main text right after the Summary of results and reads:

”The orange and green bold lines in Figure 1(c) depict experimental results of photoinduced reflectivity enhancement of Ta₂NiSe₅ single crystals obtained by time-resolved broadband THz spectroscopy as a function of frequency at temperatures of 90 K and 120 K, respectively (for details on the experimental setup, see Methods and the accompanying paper in ref. [13]).”

The results reported in the manuscript are interesting and might potentially need to be communicated to the wider specialized community, such as the readership of Nature Communications. However, upon reading the manuscript, we stumbled upon several places that prevented us from appreciating the full picture. We list these questions and suggested improvements below. Without these improvements, we will not be able to recommend the publication of the manuscript.

Our response: We thank the referee for the nice remarks about the novelty and broader implications of our work. We welcome the constructive criticism and address each point raised by the referee below.

1. Crystalline substances normally have an orderly and repetitive atomic arrangement. This notion suggests that time-crystalline behavior is similar to time crystals, where, under time translations, the period increases compared to the period of the Hamiltonian. This is exactly what is reported in the present manuscript, which makes us wonder if the suggested terminology is the best. We ask the authors to elaborate on this choice of terminology in the manuscript. In any case, it would help avoid associating new terms with the observed effect and explain its physics in simpler terms accessible to a wider community.

Our response: We thank the referee for this comment. Before we respond, we need to address an unfortunate conflict of terminology in the literature. "*Time crystals*" point to spontaneous symmetry breaking in time as mentioned by the referee. In contrast, "*photonic time crystals*" do not point to symmetry breaking at all. Instead, a photonic time crystal is a photonic crystal *in time*.

Photonic crystals are metamaterials whose effective refractive index oscillates in space modifying light propagation inside the medium. Similarly, a photonic time crystal is a metamaterial with effective refractive index oscillating in time. The main focus of the current study lies in creation of an effective medium with properties varying with time (i.e., a photonic time crystal) in the THz region due the presence of squeezed phonon oscillations.

After clarifying the subtle difference in terminology between *time crystals* and *photonic time crystals*, we can now explain the reasons for invoking the language of photonic time crystals:

1. We want to make strong connections to the very active and vibrant community which studies time-varying metamaterials. In particular, the term "photonic time crystal" was coined by a well-cited article: Lyubarov *et al.*, Amplified emission and lasing in photonic time crystals. *Science* **377**, 425-428 (2022) (DOI: 10.1126/science.abo3324, ref. [3] in our manuscript). In this respect, our work directly contributes to the existing library of literature on photonic time crystals by elucidating how strong electron-phonon coupling and squeezing oscillations of an IR-active phonon can lead to a THz photonic time crystal, and consequently to THz amplification.
2. In conjunction with the previous discussion, we hope that the Referee agrees that the term "photonic time crystal", very efficiently and elegantly, replaces the expression: *effective medium with a dielectric constant which oscillates in time*.

Changes made to the manuscript: We acknowledge the confusion that our terminology may cause despite our reasoning behind using it. To clarify further what we mean by "photonic time crystal", we modified the first paragraph of the introduction to read:

"Manipulating materials in time and space through optical pumping paves the way for new phenomena and functionalities. A notable example in this field are photonic time crystals, defined as materials whose effective dielectric properties oscillate periodically in time, in analogy to photonic crystals whose effective dielectric properties oscillate periodically in

space [1]. Photonic time-crystals have emerged as a promising platform for THz amplification, lasing [2, 3] or diffraction scattering in time [4], with potential applications in creating new THz lasers and enabling 6G wireless technology [5]. ”

Moreover, we have added a footnote which now appears as ref. [1] in the above sentence, which reads:

”[1] Photonic time crystals should not to be confused with discrete time crystals that refer to spontaneous symmetry-breaking in time.[50].”

Finally, we have changed the title to read:

”Photonic time-crystalline behaviour mediated by phonon squeezing in Ta₂NiSe₅”

We hope that the above changes eliminate any confusion that might arise from using the term ”photonic time crystal”, and again we thank the referee for bringing it to our attention.

2. The origin of eq 12 is unclear for a non-specialized reader. After screening Ref [48] of the manuscript, it is still somewhat obscure why the linear in Q coupling of the effective dipole of the phonon to the electronic dipole should have the form of Eq 12. We suggest the authors introduce more arguments and discussions supporting this particular coupling. Particularly, it would help to see the derivation of eq 12 based on a general symmetry-based scenario.

Our response: Equation (12), which now appears as Equation (3) in the modified manuscript, is exactly rooted in symmetry arguments as the Referee anticipates. In an inversion symmetric system, such as Ta₂NiSe₅, IR-active phonons are odd under inversion, $Q \xrightarrow{inv.} -Q$, while the density operator for each band is even, $c_{1,k}^\dagger c_{1,k} \xrightarrow{inv.} c_{1,k}^\dagger c_{1,k}$, which precludes a linear coupling between density operators and IR phonons. On the contrary, k -dependent transition operators between bands of opposite parity are odd under inversion, $\hat{d}_k = c_{1,k}^\dagger c_{2,k} + c_{2,k}^\dagger c_{1,k} \xrightarrow{inv.} -\hat{d}_k$, which allows for the linear coupling between d_k and Q in the Hamiltonian. The coupling λ_k simply encodes the possibility for a single phonon

mode to couple with different strength to different momentum components of the electronic transition.

Changes made to the manuscript: To reflect this clarification we have modified the paragraph before the equation to read:

”From symmetry arguments, the IR phonon coordinate is odd under inversion, and cannot couple linearly to the electron density which is even under inversion. However, it can couple linearly to electronic dipole transitions which are odd under inversion.”

3. The theory line in Figure 1c is one of the main messages of the work. We wanted to trace the details and specifics of the calculation, particularly and noticed that it was given by Fresnel eqs. 11. The matching with experiments was achieved upon fixing four parameters (as far as we could trace). The authors should comment on a) How reasonable are the fixed values of fitting parameters? b) How much the result will deviate upon changing the values of the fitting parameters.

Our response: We thank the referee for addressing the heart of our results. Below we comment on the fitting of the 4 quantities A_{drive} , ω_d , $n(\omega)$ and damping which appears as a small imaginary part of $n(\omega)$:

(i) ω_d is fixed to be 9.4 THz corresponding to squeezing fluctuations of the 4.7 THz IR mode. This is well motivated by the DFT calculations that show the 4.7 THz IR mode to be very strongly coupled to electrons. Modes that are far in frequency from 4.7 THz would show resonances at different frequencies. The fact that we capture the main resonances found in experiments makes us quite confident that this is the correct parameter.

(ii) $n(\omega)$ is fixed assuming it is purely real. This is reasonable since Ta_2NiSe_5 is an insulator therefore we would expect the real part to be larger than the imaginary part coming from conductivity. This should reliably recover the main polariton branches present in Ta_2NiSe_5 and give rise to the correct resonances. Since it is fully fitted to the experimental value of reflectivity it is very reasonable.

(iii) A_{drive} is reasonable in the sense that it corresponds only to small time-dependent changes in the permitivity compared to the equilibrium permitivity: $\frac{A_{drive}}{n(\omega)^2\omega^2/c^2} \sim 0.01 - 0.1$ in the frequency range of the THz reflectivity. This means that we can treat A_{drive}

perturbatively as we have assumed.

However, the precise value of A_{drive} is hard to determine because it sensitively depends on dissipation around that frequency. For example, if the dissipation is larger, then a larger amplitude would be needed to achieve the same amplification with qualitatively similar shape. Therefore, our uncertainty of fitting A_{drive} experimentally, comes from our uncertainty in determining correctly the dissipation in the system. Such dissipation cannot be reliably determined from first principles and experimentally it requires to know the complex reflection amplitude rather than simply the reflectivity which we don't have access to in this experiment. A careful measurement of the linear complex reflection coefficient would allow for better constraining this parameter.

Changes made to the manuscript: To make our assumptions clearer we have added in the Methods, in Theoretical fit of the parametric reflectivity amplification in pumped Ta_2NiSe_5 , the sentence:

"In particular, $\omega_d = 9.4$ THz was set to match the squeezing fluctuations of the 4.7 THz mode, which we derived from DFT as being the most strongly coupled mode to the electrons in that frequency region. Due to the resonant nature of the effect, different frequency choices would show a different pattern of resonances. The fitted A_{drive} corresponds to a small time-dependent change in permittivity compared to the equilibrium permittivity, $\frac{A_{drive}}{\omega^2 n^2(\omega)/c^2} \sim 0.01 - 0.1$, which lies within our perturbative approach.

We note in the present work as mentioned earlier in this section, it is not possible to estimate experimentally the imaginary part of the refractive index, accounting for dissipation. This leads to an ambiguity in A_{drive} , since the presence of more dissipation would need a larger drive amplitude to achieve the same amplification. We found the best fit by fitting both A_{drive} and dissipation together. Careful determination of the equilibrium complex reflectivity would allow us to better constrain this parameter from the photo-induced reflectivity."

We hope this clarifies our fitting procedure.

4. The authors described two phenomena: the 4.7 THz reflection peak under the drive frequency of 9.4 THz (photonic time crystal), and the parametric phonon oscillation at 4.7 THz, whose variance oscillates at 7.4 THZ. It is unclear to us how these two are related. It would be helpful if the authors added more discussion on this.

Our response: We apologize for the confusion. When Ta_2NiSe_5 is photo-excited, the IR-active phonon at 4.7 THz is squeezed and therefore its variance oscillates at twice the phonon frequency at 9.4 THz. Through phonon-phonon nonlinearities, oscillations of the variance modulate periodically the permittivity of material at 9.4 THz creating a photonic time crystal (dielectric medium with time-varying refractive index). The photonic time crystal can then amplify a reflected THz probe beam at 4.7 THz.

In short, the two effects are intimately related: phonon squeezing at 9.4 THz leads to a photonic time crystal at 9.4 THz through nonlinear interactions. The photonic time crystal then leads to amplification of a probe beam reflected from the interface of the material.

Changes made to the manuscript: To illuminate this connection between all the different parts, we have re-organized the manuscript considerably. In the results section, we now first discuss the electron phonon coupling, then the creation of the squeezed phonon state upon electron pumping, then we discuss amplification from a photonic time crystal which leads to amplification. Finally, we have moved a section from the methods section to the Results called: ” *Photonic time crystal from squeezing dynamics of phonons* ”. We hope that by putting this section in the Results, it is now much clearer how the squeezing phonon dynamics lead to the photonic time crystal through phonon nonlinearities.

5. A related question to point 4. In fig. 2b, there exist peaks other than the 4.7 THz. Why those peaks don’t result in peaks in the reflectivity? Other ab initio DFT parameters and results should be compared to the Fresnel equations. Otherwise, the conclusion ”the 4.7 THz mode is responsible for the parametric amplification observed in experiments” is not supported by the data.

Our response: The peaks in the IR activity plot (Fig. 2b) show the different IR active phonons around the region of interest. In principle, all IR active phonons could be squeezed and lead to amplification. However, the strength of such a non-linear process depends on the strength of the electron phonon coupling as shown by the current equation 4. From DFT calculations we show that the electron-phonon coupling for the 4.7 THz phonon to conduction bands 2 and 3 is by far the strongest as summarized in Supplementary Fig. 2. Moreover, fitting the photo-induced dielectric function of Ta_2NiSe_5 using an oscillatory term

of 9.4 THz displays an excellent theory-experiment agreement. As mentioned before, 9.4 THz is essentially the corresponding squeezing oscillation of the 4.7 THz IR-active phonon with B_{3u} symmetry. Thus, our results, in concert with the experimental data, suggest that the dominant contribution comes from the 4.7 THz phonon.

We cordially thank the referee for reviewing our manuscript and for the many suggestions that helped us improve the paper.

REVIEWER 2

Our response: We thank the referee for the valuable suggestions that helped us improve our manuscript.

REVIEWER 3

The present work deals with the amplification of THz reflectivity of an insulating material following electron photoexcitation above the gap. Experimental results are presented and a theoretical model developed, which can be used to well fit the data, and offers a physical explanation of the phenomenon. The latter entails that the photoexcited electron population induces oscillations of the phonon quadrature in a squeezed state, which in turn works as a parametric amplifier of a THz probe tuned around that phonon mode.

My opinion as a non-expert in the field of optically pumped materials is that the present observations are very interesting as well as technologically promising. The physical mechanism identified here seems plausible and generic enough to impact and spur investigations in other materials.

I am therefore in favour of publication. However, I suggest substantial reorganisation of the presentation, which I would break down in the following points:

We thank the reviewer for finding our results interesting and potentially impactful. We welcome the referee’s feedback which we address point by point below.

1. Title. The classification of the behaviour as a ”time crystal” seems not very appropriate and also unnecessary for the whole message. In my view, time crystals are interesting because the spontaneous breaking of TTI and the resulting spontaneous oscillations are infinitely long-lived or at least exceptionally robust despite the fact that one is dealing with a macroscopic number of interacting degrees of freedom, among which the energy can in principle be redistributed – leading to heating and time-crystal melting. In the present work there is no study of the oscillations-lifetime and no argument sustaining their exceptional robustness. Therefore, I suggest to modify the title to ”Theory of THz amplification mediated...”. In my opinion, pitching the work like this would be more appealing and relevant for the science of quantum materials than referring to the somewhat ”synthetic” albeit fundamentally exciting phenomenon of time crystals.

Our response: We thank the referee for pointing this out. We understand the confusion in using terminology ”time crystals” and have provided a response to a similar question raised by referee 1 that we repeat here for convenience:

Before we respond, we need to address an unfortunate conflict of terminology in the literature. ”*Time crystals*” point to spontaneous symmetry breaking in time as mentioned by the referee. In contrast, ”*photonic time crystals*” do not point to symmetry breaking at all. Instead, a photonic time crystal is a photonic crystal *in time*.

Photonic crystals are metamaterials whose effective refractive index oscillates in space modifying light propagation inside the medium. Similarly, a photonic time crystal is a metamaterial with effective refractive index oscillating in time. The main focus of the current study lies in creation of an effective medium with properties varying with time (i.e., a photonic time crystal) in the THz region due the presence of squeezed phonon oscillations.

After clarifying the subtle difference in terminology between *time crystals* and *photonic time crystals*, we can now explain the reasons for invoking the language of photonic time crystals:

1. We want to make strong connections to the very active and vibrant community which

studies time-varying metamaterials. In particular, the term "photonic time crystal" was coined by a well-cited article: Lyubarov *et al.*, Amplified emission and lasing in photonic time crystals. *Science* **377**, 425-428 (2022) (DOI: 10.1126/science.abo3324, ref. [3] in our manuscript). In this respect, our work directly contributes to the existing library of literature on photonic time crystals by elucidating how strong electron-phonon coupling and squeezing oscillations of an IR-active phonon can lead to a THz photonic time crystal, and consequently to THz amplification.

2. In conjunction with the previous discussion, we hope that the Referee agrees that the term "photonic time crystal", very efficiently and elegantly, replaces the expression: *effective medium with a dielectric constant which oscillates in time.*

Changes made to the manuscript: We acknowledge the confusion that our terminology may cause despite our reasoning behind using it. To clarify further what we mean by "photonic time crystal", we modified the first paragraph of the introduction to read:

"Manipulating materials in time and space through optical pumping paves the way for new phenomena and functionalities. A notable example in this field are photonic time crystals, defined as materials whose effective dielectric properties oscillate periodically in time, in analogy to photonic crystals whose effective dielectric properties oscillate periodically in space [1]. Photonic time-crystals have emerged as a promising platform for THz amplification, lasing [2, 3] or diffraction scattering in time [4], with potential applications in creating new THz lasers and enabling 6G wireless technology [5]."

Moreover, we have added a footnote which now appears as reference [1] in the above sentence, which reads:

"[1] Photonic time crystals should not to be confused with discrete time crystals that refer to spontaneous symmetry-breaking in time.[50]."

Finally, we have changed the title to read:

"Photonic time-crystalline behaviour mediated by phonon squeezing in Ta₂NiSe₅"

We hope that the above changes eliminate any confusion that might arise from using the term "photonic time crystal", and again we thank the referee for bringing it to our attention.

With the above explanation, we sincerely hope that the referee will be convinced that the terminology as is used in the current form is both appropriate and useful in communicating our results efficiently.

2. Experimental part. I understand that the experimental data is not published anywhere and thus the present manuscript should be regarded as a theory-experiment collaboration. If this is the case, then I find that too little relevance is given to the experimental part, which provides the starting point and orientation for the theory part. The title is clearly classifying this work as a theory work, and in the main text the description of the experimental methods is maybe 5% of the whole, without even a dedicated section.

Our response: We thank the referee for the comment. The experimental paper which accompanies our work, has now been published in *Nature Materials* (DOI: <https://doi.org/10.1038/s41563-023-01755-2>, ref. [13] in our manuscript). In the experimental paper, we highlight how nonlinear processes such as electron-phonon mediated amplification of reflectivity can be used as a reporter for the excitonic order parameter in Ta_2NiSe_5 , by focusing on the peculiar temperature dependence of the result. The theory of the amplification there is presented on a surface level.

Conversely, in this paper, our primary focus is on advancing the theoretical framework, elucidating how a robust electron-phonon coupling can introduce a novel mechanism for generating low-frequency THz photonic time crystals by high-frequency optical pumping. From this perspective, the theory uncovers a universal phenomenon, with experiments on Ta_2NiSe_5 providing a compelling illustration of the effect due to its pronounced electron-phonon coupling.

Changes made to the manuscript: As per the referee suggested, we now made some modifications in the manuscript and added a new section titled "Experimental details" in the Methods section of our manuscript that provides a thorough account of the data acquisition scheme. We point to this change in the main text right after the Summary of results section, which reads:

”The orange and green bold lines in Fig. 1c depict experimental results of photoinduced reflectivity enhancement of Ta₂NiSe₅ single crystals obtained by time-resolved broadband THz spectroscopy as a function of frequency at temperatures of 90 K and 120 K, respectively (for details on the experimental setup, see Methods and the accompanying paper in ref. [13]).”

We hope the above changes would make the manuscript self-contained and clarify our works connection to the experimental paper that is already published.

3. Explanation of the theory. I find the explanation of the model and calculations not easy to follow. This is probably in part caused by my limited expertise in the field, but I find that in a broad-audience journal the basic logic, ingredients, and assumptions should become clear to all readers, and from the main text. Therefore, I suggest some reorganisation. The main suggestion is to present the section ”Phonon-squeezing initiated ... electrons” BEFORE the section ”Parametric amplification ... Ta₂NiSe₅”. The reason is that I understood the parametric-amplification model (signal-idler mixing) only after I read the Phonon-squeezing section, and in particular the derivation in the METHODS. In particular, I think that few equations from the METHODS-section ”Floquet matter from...phonons” need to be moved to the main text, e.g. the full Hamiltonian (39), and eq.(40) or its linearised version, otherwise it is really not clear how the interplay between E-field and phonons works and is modelled. Starting from (39) one could also derive, at the same time, equation (40), describing the coupling of $\langle Q \rangle$ to the THz probe-field and to the quadrature $\langle Q^2 \rangle$ through anharmonicity, as well as equation (38), describing the excitation of the quadrature oscillations from the electron population. This would result in a much more organised and compact presentation, the crucial steps of which can be put in the main text as suggested above.

Our response We thank the referee for the suggestions. After giving a thorough read, we agree with the points mentioned by the referee and have made significant changes accordingly.

Changes made to the manuscript: We moved the ”Phonon squeezing initiated by photoexcited electrons” subsection before the ”Parametric amplification of reflectivity in pumped Ta₂NiSe₅” subsection as the referee suggested. We also moved the ”Floquet matter

from squeezing dynamics of phonons” subsection from the Methods into the main text after the “Floquet-Fresnel equations” subsection and combined it with a previous section called “Photonic Time crystal”. The new section is called “**Photonic time crystal** from squeezing dynamics of phonons”.

At the same time the introduction was re-organized to reflect the new presentation.

We thank the referee for this very useful feedback and we hope that the above reorganization has improved the readability, and will make the story more vivid to the broad readership.

4. **Extent of the ab-initio support.** The abstract might sound like the whole theory model is supported by ab-initio calculations. By reading the corresponding section it is clear that the ab-initio part nicely justifies the electron-phonon coupling Hamiltonian, but not the whole subsequent calculations down to the final prediction about amplification. While the latter seem compelling and sound to me, it might be desirable to state more clearly whether and which important assumptions/simplifications are made which are not justified from the ab-initio and would in principle need confirmation through e.g. some kind of ab-initio dynamical simulations.

Our response: we completely agree with the referee’s assessment of our work. We have made changes in the abstract and in the relevant section to more accurately state the degree to which ab initio calculations support our minimal model.

Changes made to the manuscript:

In the abstract we have modified the sentence to read:

“...simplified electron-phonon Hamiltonian motivated by ab-initio DFT calculations.”

To make the claim that our theory is supported by ab initio calculations softer.

Moreover, in the ab-initio section we make clear what parameters are not extracted from first principles:

“ Therefore, the DFT calculation provides strong support for the minimal electron-phonon model and uncovers the dominant electronic bands and phonon involved in the nonlinear process. The other ingredients needed to extract the amplification amplitude, i.e., phonon non-linearities, phonon dissipation and the dynamics of the photo-excited electrons responsible for phonon squeezing, were included phenomenologically as described in the photo-induce reflectivity fitting section.”

5. Fitting. It was not clear to me which parameters are fitted and which obtained otherwise. Even if briefly, this seems important to be mentioned in the main text, and reiterated in the detailed derivation of the methods section.

Our response: We thank the referee for their remark. We have now added a more nuanced discussion on what it was fitted or obtained otherwise by adding the sentence in Methods in the theoretical fit section.

Changes made to the manuscript: We added the following discussion:

"In particular, $\omega_d = 9.4$ THz was set to match the squeezing fluctuations of the 4.7 THz mode, which we derived from DFT as being the most strongly coupled mode to the electrons in that frequency region. Due to the resonant nature of the effect, different frequency choices would show a different pattern of resonances. The fitted A_{drive} corresponds to a small time-dependent change in permittivity compared to the equilibrium permittivity, $\frac{A_{\text{drive}}}{\omega^2 n^2(\omega)/c^2} \sim 0.01 - 0.1$, which lies within our perturbative approach.

We note in the present work as mentioned earlier in this section, it is not possible to estimate experimentally the imaginary part of the refractive index, accounting for dissipation. This leads to an ambiguity in A_{drive} , since the presence of more dissipation would need a larger drive amplitude to achieve the same amplification. We found the best fit by fitting both A_{drive} and dissipation together. Careful determination of the equilibrium complex reflectivity would allow us to better constrain this parameter from the photoinduced reflectivity."

At the same time as per the Referee's suggestion, we have added in the main text:

"To fit the data, we choose a drive at 9.4 THz motivated by the DFT calculations outlined in the ab-initio calculations section. We then use equation (13) and (17) and fit the parameters A_{drive} and $n(\omega)$ to the experimental data allowing for small changes in the static properties of the system such as a photoinduced conductivity stemming from photoexcited charge carriers. The fitted parameters are given in the Methods section."

We now hope the new additions make our fitting procedure clear.

6. Further more detailed remarks:

a. Eq.(2). This equation seems a bit disconnected from the formalism developed later in the text.

Our response: We thank the referee for this remark. Eq. (2) is the prototypical equation of a parametrically driven oscillator, and it is meant to give intuition behind the parametric amplification process. In particular, it captures the parametric resonant condition that is somehow hidden in the subsequent discussion based on equations of motion of the electric field.

Changes made to the manuscript: We try to clarify what we are trying to communicate with this term by rewording the paragraph as :

”The parametric amplification of reflectivity in the photonic time crystal created by the squeezing phonon oscillations can be understood as follows. The oscillations of phonon fluctuations, parametrically drive the rest of the material through nonlinear interactions, which can be schematically represented as:

$$H_{\text{ampl.}} = X(t)a_{1,k}^\dagger a_{2,-k}^\dagger + h.c.,$$

where $X(t)$ is the squeezing induced drive while $\{a_{1,k}^\dagger, a_{2,-k}^\dagger\}$ are the creation operators of photons with opposite momentum which may be associated with different phonon-polariton bands $\{1, 2\}$ at frequencies parametrically resonant with the drive, $\omega_1(k) + \omega_2(-k) = \omega_d$.”

b. Treatment of the E-field. The E-field enters both eq.(35) and eq.(40). However, it is treated differently between the two, suggesting that two different sets of modes are relevant for these equations. It seems that in (35) only the $k=0$ mode is considered (not excited by the probe), while in (40) the signal-idler modes at finite k are considered. Some further discussion would help.

Our response: Indeed, the Referee is correct in pointing out that we have suppressed the momentum labeling in the two cases. We have now remedied the situation and make clear which modes we are discussing at each section. Moreover, we implicitly assume in our paper that only the $k = 0$ squeezing dynamics are excited by the electrons, which we make clear in the current version of the manuscript.

Changes made to manuscript: To clarify the implicit assumption in our treatment we have added at the end of the methods section the paragraph:

”In our treatment, we implicitly assume that only $k = 0$ phonons are excited by photoelectrons. The contribution of finite momentum squeezing modes is beyond the scope of our work.”

Moreover, in the main text we explicitly discriminate between $k = 0$ and finite k modes to avoid confusion. Current equations (18) - (25) have been edited to reflect this change.

c. From (39) to (40) some damping γ appears without discussion it seems. Please elaborate.

Our response: We thank the referee for the comment. We have added comments about the meaning of gammas.

Changes made to the manuscript: We have made the following addition in the text:

” To describe the dynamics of phonons, we include phenomenologically the damping term γ , which captures the decay of the phonon to other degrees of freedom. ”

d. In eqs.(40,41) the driving function f has disappeared. Why? I guess it comes from the linearisation as done in the eqs. for $\langle Q^2 \rangle$, but then f should still be there in (40). Please explain.

Our response: We thank the referee for their remark and explain our choice in the text.

Changes made to the manuscript: The following text is added:

” The contribution from $f(t)$ corresponds to fast dynamics of the photoelectrons and therefore has a broad spectrum. As such, it does not lead to resonant dynamics and will at most contribute as an incoherent background. In contrast, squeezing oscillations oscillating at a specific frequency can lead to parametric amplification that is narrowband. Hence, we drop the contribution of $f(t)$ and focus on the consequences of the squeezing mode oscillations. ”

We thank the Referee for their many helpful remarks that we think have considerably improved the flow and readability of our work.

REVIEWERS' COMMENTS

Reviewer #1 (Remarks to the Author):

We have read the current version of the manuscript and the responses to the referee's comments. The authors satisfactorily addressed all our points in the report. We can now recommend the publication of the manuscript in Nature Communications.

Reviewer #2 (Remarks to the Author):

Reviewer #3 (Remarks to the Author):

In their reply and in the new version of the manuscript the Authors satisfactorily addressed the Referees remarks and suggestions.

I am thus happy to recommend the manuscript for publication.